# Novel Enhanced Mammalian Cell Transient Expression Vector via Promoter Combination

**DOI:** 10.3390/ijms25042330

**Published:** 2024-02-16

**Authors:** SunKyung Yoon, SeJin Park, JuneWoo Lee, Byoungguk Kim, WonSeok Gwak

**Affiliations:** Division of Clinical Vaccine Research, Center for Vaccine Research, National Institute of Infectious Diseases, National Institute of Health, Korea Disease Control and Prevention Agency, Cheongju 28160, Chungcheongbuk-do, Republic of Korea; skyoon01@korea.kr (S.Y.); sejin27@korea.kr (S.P.); junewoo1213@korea.kr (J.L.); kimbg10@korea.kr (B.K.)

**Keywords:** transient expression vector, additional promoter, antigen, cytomegalovirus promoter, human elongation factor 1-alpha promoter

## Abstract

During the emergence of infectious diseases, evaluating the efficacy of newly developed vaccines requires antigen proteins. Available methods enhance antigen protein productivity; however, structural modifications may occur. Therefore, we aimed to construct a novel transient overexpression vector capable of rapidly producing large quantities of antigenic proteins in mammalian cell lines. This involved expanding beyond the exclusive use of the human cytomegalovirus (CMV) promoter, and was achieved by incorporating a transcriptional enhancer (CMV enhancer), a translational enhancer (woodchuck hepatitis virus post-transcriptional regulatory element), and a promoter based on the CMV promoter. Twenty novel transient expression vectors were constructed, with the vector containing the human elongation factor 1-alpha (EF-1a) promoter showing the highest efficiency in expressing foreign proteins. This vector exhibited an approximately 27-fold higher expression of enhanced green fluorescent protein than the control vector containing only the CMV promoter. It also expressed the highest level of severe acute respiratory syndrome coronavirus 2 receptor-binding domain protein. These observations possibly result from the simultaneous enhancement of the transcriptional activity of the CMV promoter and the human EF-1a promoter by the CMV enhancer. Additionally, the synergistic effect between the CMV and human EF-1a promoters likely contributed to the further enhancement of protein expression.

## 1. Introduction

Antigen proteins are essential in assessing the efficacy of vaccines during clinical trials and in antibody production. Their applications are diverse, extending to direct use as vaccines [1]. Traditional vaccine development is a time-consuming process, requiring approximately 10 years or more to progress from candidate material development to the initiation of clinical trials [2]. However, in response to the urgent need posed by coronavirus disease (COVID-19), novel vaccine platforms, including nucleic acid vaccines (such as DNA and RNA), viral vector vaccines, and subunit vaccines, have been explored. This has led to a significant reduction in the time taken for vaccine development, with clinical trials starting within 70 days of the outbreak [3,4]. This accelerated speed means that antigen proteins must also be provided quickly to assess the efficacy of newly developed vaccines against the emergence of infectious diseases in the future.

However, it is also essential to select a suitable recombinant protein expression system. This is because a protein with a very high similarity to the antigen of the target pathogen is required to perform an accurate efficacy assessment. Considerations of post-translational modifications, such as folding and glycosylation of antigen proteins, are essential [5,6]. Antigen proteins used for evaluating vaccine efficacy are almost membrane proteins, which are poorly expressed and difficult to mass-produce [7]. *E. coli* or yeast cells offer a faster and more cost-effective route for recombinant protein production compared to other expression systems [8,9,10]. However, there is a notable risk of the expressed antigen protein adopting an incorrect structure [11]. Expression systems using insect cells and plants support post-translational modifications that are superior to that of *E. coli* by virtue of using eukaryotic cells; however, they are still inferior to those of mammals, which are higher organisms [12,13,14]. Transgenic insect cell lines capable of post-translation modification of mammalian cells have been developed, but they are still incomplete [13]. On the contrary, mammalian cells support transformation well after translation. However, their productivity is low, making it difficult to respond quickly to emerging infectious diseases, and their production costs are high [11,14,15,16]. Mammalian cells can also produce antigen proteins by generating transgenic cell lines. However, the cell line generation and screening process is complex and time-consuming. Mammalian transient expression is capable of rapid production but low yield.

Therefore, the development of new overexpression vectors is necessary to facilitate the rapid mass production of recombinant proteins in mammalian cells [17]. Various methods have been employed to increase the productivity of antigen proteins, including the use of fusion partners, extracellular secretion, and expression time control [18,19]. However, using fusion partners to improve stability or signal peptides for secretion may inadvertently lead to structural modifications [20]. For accurate efficacy evaluation, the antigen protein must undergo mass expression in its unique form; therefore, in this study, we aimed to construct a vector with increased transcriptional activity through the combination of a promoter and a transcription-enhancing factor [21]. To achieve this, we planned an overexpression vector with a novel structure capable of expressing one gene as two promoters. One of the two promoters was the cytomegalovirus (CMV) promoter commonly used in mammalian cell expression systems, with five types of additional promoters placed in its upstream region. In addition, to further increase the expression efficiency, the woodchuck hepatitis virus post-transcriptional regulatory element (WPRE) of the CMV encoder and woodchuck hepatitis virus was added. Twenty combinations of overexpression vectors were established, and the expression levels of foreign proteins were compared and evaluated to select the most efficient combination of overexpression vectors.

## 2. Results

### 2.1. Construction of Mammalian Overexpression Vectors

Novel expression vectors with increased expression efficiency based on the CMV promoter were developed by incorporating the CMV enhancer, WPRE, and additional promoters. The functions of the CMV enhancer and WPRE used in this study and the activities of additional promoters have been previously characterized. The CMV enhancer and WPRE were used as transcriptional or translational enhancers. In contrast, chicken B-actin, human elongation factor 1-alpha (EF-1a), phosphoglycerate kinase (PGK), gibbon ape leukemia virus U3 (GALV U3), and actin beta (ACTB) promoters were used as additional promoters. According to previous reports, the CMV enhancer is located upstream of the promoter, whereas the WPRE sequences are located downstream of the target protein gene. Expression vectors across 4 groups (a total of 24 vectors, including control vectors for each group) were constructed using each additional promoter in an expression vector containing a transcription or translation enhancer along with a CMV promoter (Figure 1). All overexpression vectors consisted of the same sequence, except for the type of additional promoter, presence or absence of an enhancer, and WPRE.

The expression efficiency of these vectors was evaluated using enhanced green fluorescent protein (EGFP) expression. For comparative analysis among the overexpression vectors, EGFP expression was analyzed by measuring the fluorescence intensity.

### 2.2. Determination of the Optimal Expression Time Point

After the expression of EGFP, the expression level of the overexpression vectors was evaluated by comparing the fluorescence intensity. Before that, it was necessary to confirm when EGFP was most expressed in human embryonic kidney 293T (HEK293T) cells where the overexpression vector was transfected. HEK293T cells were transfected with an EGFP overexpression vector. To compare EGFP expression at each time point, fluorescence microscopy observations and fluorescence intensity measurements were performed at 24 h intervals up to 120 h. Under fluorescence microscopy, the expression of the fluorescent protein was observed approximately 24 h post-transfection, rapidly increasing after 48 h (Figure 2A). Cells were collected at each observation time point, and fluorescence intensity was measured using the expressed EGFP. The fluorescence intensity was highest at 72 h post-transfection and then decreased (Figure 2B). Accordingly, the optimal time point for comparative evaluation of the expression levels of overexpression vectors was determined to be 72 h post-transfection.

### 2.3. Comparative Evaluation of Overexpression Vectors through EGFP Expression

To systematically compare the performance of the twenty types of overexpression vectors distributed across the four groups through EGFP expression, the same amount of overexpression vector plasmid was transfected under the same conditions. Subsequently, observation using a fluorescence microscope and measurement of fluorescence intensity were performed. Fluorescence microscopy analysis revealed that in Group #delta, which lacked the CMV enhancer and WPRE, all vectors except pAC exhibited fluorescent protein expression patterns similar to those of the control pC vector. Group #1, with an additional CMV enhancer compared with group #delta, showed the strongest fluorescent protein expression with the pCEC vector, and the other vectors (pCAC, pCCC, pCGC, and pCPC) were similar to the control pCC. Group #2, with an additional WPRE compared with group #delta, showed fluorescent protein expression similar to that of control pC-W in all vectors, and none exhibited a noticeably high expression level like those in Group #1. Group #3, with an additional CMV enhancer and WPRE, showed the strongest fluorescent protein expression in the pCEC-W vector, and the other vectors (pCAC-W, pCCC-W, pCGC-W, and pCPC-W) were similar to the control pCC-W (Figure 3A). Fluorescence measurements were performed to quantitatively compare the expression intensities of the overexpression vectors (Figure 3B–E). The overexpression vectors (excluding pCGC in Group #1) across all groups consistently demonstrated higher expression intensity than the control vectors of each group (Figure 3B–E). In Group #delta, all overexpression vectors showed higher fluorescence intensity than the control pC. The expression level of EGFP was increased by an auxiliary promoter placed upstream of the CMV promoter. Among them, the pAC vector exhibited the highest expression level, approximately three times that of the control vector and twice that of the remaining overexpression vectors (Figure 3B). In Group #1, the pCEC vector displayed the highest expression level, approximately 27 times that of the control vector. The expression level of the human elongation factor 1-alpha (EF1-a) promoter was elevated by the CMV enhancer. Other additional promoters showed no effect or the expression level was slightly lowered (Figure 3C). In Group #2, the pAC-W vector exhibited the highest expression level, approximately three times that of the control vector. However, compared with Group #delta, all promoters exhibited reduced expression levels by WPRE. Among them, the human EF1-a promoter showed the least impact, and the rest were reduced five- to six-fold (Figure 3D). Finally, in Group #3, the level of pCEC-W vector expression was approximately 115 times stronger than that of the control vector. However, compared with Group #1, all promoters exhibited reduced expression levels by WPRE. Of these, the human EF1-a promoter showed the least impact, and the rest were reduced three- to six-fold (Figure 3E). Overexpression vectors exhibited significantly different expression levels of EGFP depending on the type of additional promoter or the presence or absence of CMV enhancer or WPRE (Figure 3). Among them, the pCEC and pCEC-W vectors, which showed significantly higher EGFP expression levels, are expected to facilitate the mass production of antigen proteins.

### 2.4. Application of Antigenic Protein Expression of Overexpression Vectors

Expression levels of all overexpression vectors were assessed using EGFP. Among them, the pCEC vector showed a superior level of EGFP expression. Therefore, it was necessary to further verify that the pCEC vector also expresses antigenic proteins other than EGFP at a high level. The antigen protein was the severe acute respiratory syndrome coronavirus 2 (SARS-CoV-2) receptor-binding domain (RBD), and all vectors of Group #1, including the pCEC vector, were used. To create a recombinant overexpression vector that overexpressed the SARS-CoV-2 receptor-binding domain (RBD), six vectors in Group #1 were treated with the restriction enzymes EcoRI and EcoRV to remove the EGFP gene and facilitate the cloning of the SARS-CoV-2 RBD gene. The resulting SARS-CoV-2 RBD gene overexpression vectors produced by gene cloning were validated by DNA electrophoresis after treatment with specific restriction enzymes. To confirm the overexpression of the SARS-CoV-2 RBD protein, HEK293T cells were transfected with a recombinant overexpression vector. Seventy-two hours post-transfection, cells were collected, lysed with M-PER lysis buffer, and electrophoresed on a 4–20% gradient sodium dodecyl sulfate-polyacrylamide gel electrophoresis (SDS-PAGE) gel. The separated proteins were subsequently transferred to a polyvinylidene fluoride (PVDF) membrane and treated with an anti-RBD antibody specific to SARS-CoV-2 RBD. Western blotting images were captured using a chemiluminescence imaging system (ChemiDoc MP, Bio-Rad, Hercules, CA, USA). The outcomes revealed a strong signal corresponding to the approximately 29-kDa RBD protein only in the pCEC vector that exhibited the highest EGFP expression; expression could not be confirmed in the remaining vectors, including the control vector (Figure 4).

Comparison of SARS-CoV-2 RBD protein expression levels using vectors from Group 1, which included the pCEC vector with the highest enhanced green fluorescent protein (EGFP) expression level.

## 3. Discussion

The pandemic of infectious diseases, including COVID-19, caused enormous social and economic damage. In fact, the SARS outbreak in 2002, the swine flu pandemic seven years later in 2009, the MERS outbreak six years later in 2015, and the COVID-19 outbreak five years later in 2020 continued to shorten the cycle of new infectious diseases. In addition, the speed of vaccine development to cope with emerging infectious diseases is also accelerating. Therefore, antigen proteins for accurate efficacy evaluation of newly developed vaccines must also be provided quickly, and the development of an overexpression vector with high-quality antigen productivity is needed.

In this study, 20 new types of vectors differentiated from existing expression vectors were produced by combining promoters, CMV enhancers, and WPRE used for protein expression in mammalian cells. The expression levels of the overexpression vector were compared using EGFP and verified using the SARS-CoV-2 RBD antigen protein. The fluorescence intensity of EGFP was used to compare the expression levels across twenty overexpression vectors distributed in four groups (Figure 3B–E). In Groups #delta and #2, where the CMV enhancer was absent, the pAC vector exhibited the highest expression level, with the remaining vectors exhibiting similar expression levels. The expression of all overexpression vectors was higher than that of the pC and pC-W vectors, which were controls in each group. Thus, it appears that the additional promoter influenced the increase in the expression level of EGFP (Figure 3B,D). For protein overexpression, various strategies have been developed. Two typical methods in which two overlapping expression cassettes are placed on one plasmid or one ORF is expressed as a dual promoter. To express one ORF with dual promoters simultaneously, it is difficult for the main promoter to arrange without interfering with the transcriptional activity of the additional promoter. However, it can be a way to develop an excellent overexpression vector, as in this study, and related studies have already been reported in Bacillus and insect cells [21,22]. The basic concept of the dualpromoter system is quite intuitive, reminiscent of a high-speed train with two locomotives connected in sequence. This configuration enhances propulsion and speed [23]. In Groups #1 and #3, which contained the CMV enhancer, notable expression of the pCEC and pCEC-W vectors was observed (pEC: 17 times and pEC-W: 37 times compared to the vector without CMV enhancer) (Figure 3B–E). Previous studies have indicated that the CMV enhancer increases the expression of the EF-1a promoter, surpassing its impact on the simple EF-1a promoter by a factor of two. However, the results of this study surpass the magnitude reported in previous studies [24]. Previous studies used only EF-1a promoters, and this study used EF-1a and CMV promoters in duplicate. The higher efficacy of the CMV enhancer observed in this study is attributed not only to the enhanced transcriptional activity of the EF-1a promoter by the CMV enhancer, but also to the synergistic effect between the EF-1a and CMV promoters.

WPRE aids in the nuclear export of mRNA and forms a 3D structure that enhances the transcript expression. Additionally, WPRE prevents read-through of the poly(A) site, promotes RNA processing and maturation, and increases RNA nuclear export. WPRE plays a crucial role in enhancing the efficiency of viral vectors used for gene delivery. This sequence is used primarily in molecular biology to enhance the expression of genes delivered by viral vectors [25,26,27]. Therefore, we aimed to maximize the performance of the overexpression vector by increasing the transcription level using the CMV enhancer and increasing the translation efficiency by WPRE. However, in contrast to expectation, WPRE played a role in inhibiting the expression level of all overexpression vectors in this study. When all 24 vectors, including control vectors, were sorted based on EGFP expression levels, vectors containing WPRE, except pCEC-W, were predominantly distributed at the bottom (Figure 3B–E). In the comparison between Group #delta and Group #2, a decrease in expression level, approximately three to four times, was observed (Figure 3B,D). Similarly, in the comparison between Group #1 and Group #3, the expression levels decreased by approximately one to six times (Figure 3C,E). Previous studies have shown the inhibition of protein expression by WPRE in specific cell lines [28]. It was also reported that the expression level of EGFP was reduced by approximately 50% by WPRE in primary murine hematopoietic cells infected with the retroviral vector [29]. In this study, expression levels were also reduced by approximately 65%–85% by WPRE, depending on the vector (Figure 3B–E). Further investigations are required to determine whether the decrease in protein expression caused by WPRE occurs at the translational or transcriptional level.

Upon overexpressing the SARS-CoV-2 RBD protein using an overexpression vector, specific Western blotting analysis confirmed a strong signal exclusively in the pCEC vector (Figure 4). These findings suggest that the pCEC vector increased the mRNA transcription of the RBD protein gene, thereby increasing the protein expression level. However, the absence of RBD expression in the remaining vectors is likely linked to the stability of the protein post-mRNA transcription and translation. All overexpression vectors were transfected into HEK293T cells under the same conditions. In the expression of EGFP, the expression level differed from vector to vector, but all expressions were confirmed. This demonstrates that transcription of the EGFP gene was performed in all overexpression vectors. Thus, the expression of the SARS-CoV-2 RBD protein is also at different levels, as is the expression of EGFP, but all would have been transcribed and translated. The SARS-CoV-2 Spike protein comprises an extracellular N-terminal domain, a transmembrane domain anchored to the membrane, and an intracellular C-terminal segment [30,31]. Inherently unstable membrane proteins present challenges in structural and functional research [32]. The RBD protein is a small domain belonging to the S1 domain of the SARS-CoV-2 Spike protein. Although initially a transmembrane protein, it is designed to express only RBD and exists in the cytoplasm post-expression. It is presumed that transcription of the RBD protein gene occurred in all vectors. However, except for the pCEC vector, the transcriptional output of the remaining vectors was low, resulting in a correspondingly low expression level of the RBD protein. Moreover, it is assumed that they were degraded owing to their presence in the cytoplasm and low stability. Therefore, specific bands were not identifiable in the Western blotting analysis.

In conclusion, the pCEC and pCEC-W vectors, employing EF-1a as an additional promoter, exhibited the highest expression levels for both EGFP and SARS-CoV-2 RBD proteins. This outcome can be attributed to the CMV enhancer, serving as a CRE, which not only stimulated both the EF-1a and CMV promoters, but also potentially enhanced expression through the synergistic effect between the EF-1a and CMV promoters.

Therefore, the overexpression vector developed in this study will be a useful tool for rapid vaccine development upon the emergence of emerging infectious diseases. It is expected that the antigen protein supplied quickly can be used for accurate efficacy evaluation of newly developed vaccines and can also be used as diagnostic antigens. In addition, it is expected that production at a lower price than before will be possible by increasing productivity in various protein drug industries.

## 4. Materials and Methods

### 4.1. Cells

HEK293T cells (ATCC CRL-3216) were grown in Dulbecco’s modified Eagle’s high glucose medium (11995-065, Thermo Fisher Scientific, Waltham, MA, USA) supplemented with 10% fetal bovine serum (10082-147, Thermo Fisher Scientific, Waltham, MA, USA) and 1% penicillin/streptomycin (15140-122, Thermo Fisher Scientific, Waltham, MA, USA) at 37 °C in a humidified incubator at 5% carbon dioxide. Routine cell culture maintenance and seeding were performed according to published procedures [33]. Cells were subcultured for at least 3 passages after thawing, and did not exceed a maximum of 20 passages.

### 4.2. Plasmid Transfection

Endotoxin-free plasmid DNA was prepared using the Plasmid Midiprep Kit (12943, QIAGEN, Hilden, Germany). For overexpression of recombinant protein in HEK293T cells, 5 μg of recombinant plasmid DNA was transfected into 0.7 × 10^6^ cells per well of 6-well Clear TC-treated Multiple Well Plates (CLS3516, Corning, NY, USA) using 5 μL of Lipofectamine 3000 transfection Reagent™ and 2.5 μL of p3000 Reagent (L3000001, Invitrogen, Waltham, MA, USA), following the manufacturer’s instructions.

### 4.3. Fluorescence Observation and Intensity Measurement

The expression of EGFP in the transfected cells was observed using a fluorescence microscope (Ts2R-FL, Nikon, Minato, Japan) for 120 h at 24 h intervals. Fluorescence intensity measurements were conducted by harvesting transfected cells at 24 h intervals from 24 h to 120 h. Subsequently, cells were washed with ice-cold phosphate-buffered saline (PBS) (P2201-050, Gendepot, Katy, TX, USA), and lysates were prepared by incubating the cells with 1 mL of M-PER lysis buffer (78501, Thermo Fisher Scientific, Waltham, MA, USA) for 30 min on ice. Fluorescence was measured at room temperature using a multimode microplate reader (SpectraMax i3x; Molecular Devices, San Jose, CA, USA), following the manufacturer’s instructions from at least three independent biological replicates.

### 4.4. Selection of Expression Elements and Establishment of a Vector Construction Plan

In this study, we aimed to construct a vector with increased transcriptional activity using a strategic combination of promoters. We selected promoters, transcription enhancers, and translation enhancers with proven efficacy in previous studies (Table 1). The selected promoters included CMV [34], chicken B-actin [35], EF-1a [36], PGK [37], GALV U3 [38], and ACTB promoters [39]. CMV promoters are most commonly used for foreign protein expression in mammalian cells. Therefore, the CMV promoter was placed as the main promoter among the two promoters of the novel expression vectors. Five novel expression vectors were planned by placing five other promoters (chicken B-actin, EF-1a, PGK, GALV U3, and ACTB promoters) upstream of the CMV promoter as additional promoters (Figure 1A). In addition, as a Cis-regulatory acting element, we added a group that placed CMV enhancers in the upstream region of promoters (Figure 1B) [40], a group that placed WPRE in the downstream region of antigen protein genes as translation enhancers (Figure 1C) [25], and a group that placed both CMV enhancers and WPRE (Figure 1D). Four vector groups were created, and 24 vectors were constructed, including controls that did not have an additional promoter per group (Figure 1).

### 4.5. Construction of Expression Vectors

According to the final model for overexpression vector construction, the sequences of Group #1 vectors were synthesized (Figure 1B). Group #1 recombinant plasmids were treated with the restriction enzymes EcoR V (R3195L, New England Biolabs, Ipswich, MA, USA)/Bgl II (R0144L, New England Biolabs, Ipswich, MA, USA), and WPRE was subcloned to produce Group #3 recombinant plasmids (Figure 1D). Groups #1 and #3 recombinant plasmids were treated with restriction enzymes Hinc II (R0103L, New England Biolabs, Ipswich, MA, USA)/SnaB I (R0130L, New England Biolabs, Ipswich, MA, USA). Subsequently, they were self-ligated and transformed to produce Group #delta (Figure 1A) and Group #2 (Figure 1C), with the CMV enhancer removed.

### 4.6. Western Blotting Analysis

For Western blotting analysis, HEK293T cells were transfected with a recombinant plasmid expressing SARS-CoV-2 RBD. Transfected cells were collected 72 h post-transfection, washed with ice-cold PBS, and lysed with 300 μL of M-PER lysis buffer for 30 min on ice. The 80 μL of lysates were mixed with a 20 μL of 5× protein sample buffer, boiled for 5 min, subjected to 4–20% gradient SDS-PAGE (4561094, Bio-Rad, Hercules, CA, USA), and transferred to a 0.2 µm PVDF membrane (1704156, Bio-Rad, Hercules, CA, USA). For the SARS-CoV-2 RBD positive control, Recombinant SARS-CoV-2 Spike RBD (His-tag) (10500-CV, R&D Systems, Minneapolis, MN, USA) was purchased and used at a concentration of 100 ng. The membrane was blocked using EveryBlot blocking buffer (12010020, Bio-Rad, Hercules, CA, USA) and probed with a SARS-CoV-2 RBD (E7B3E) rabbit monoclonal antibody (63847S, Cell Signaling Technology, Danvers, MA, USA). Subsequently, the membrane was incubated with a horseradish peroxidase-conjugated anti-rabbit IgG antibody (7074S, Cell Signaling Technology, Danvers, MA, USA). The bound antibodies were detected using a chemiluminescence imaging system (ChemiDoc MP, Bio-Rad, Hercules, CA, USA), following the manufacturer’s instructions, and analyzed using Image Lab Touch Software version 3.0.1.14 (Bio-Rad, Hercules, CA, USA).

### 4.7. Statistical Analysis

Statistical analysis was performed using GraphPad Prism software, version 9.5.1 (GraphPad Software, Boston, MA, USA). Student’s *t*-test was used to evaluate statistically significant differences between the different treatment groups. A *p*-value < 0.05, *p* < 0.005, *p* < 0.0005, and *p* < 0.0001 indicated a significant difference represented by “*”, “**”, “***”, and “****”, respectively. Data are presented as the mean ± SD from at least three independent biological replicates.

## Figures and Tables

**Figure 1 ijms-25-02330-f001:**
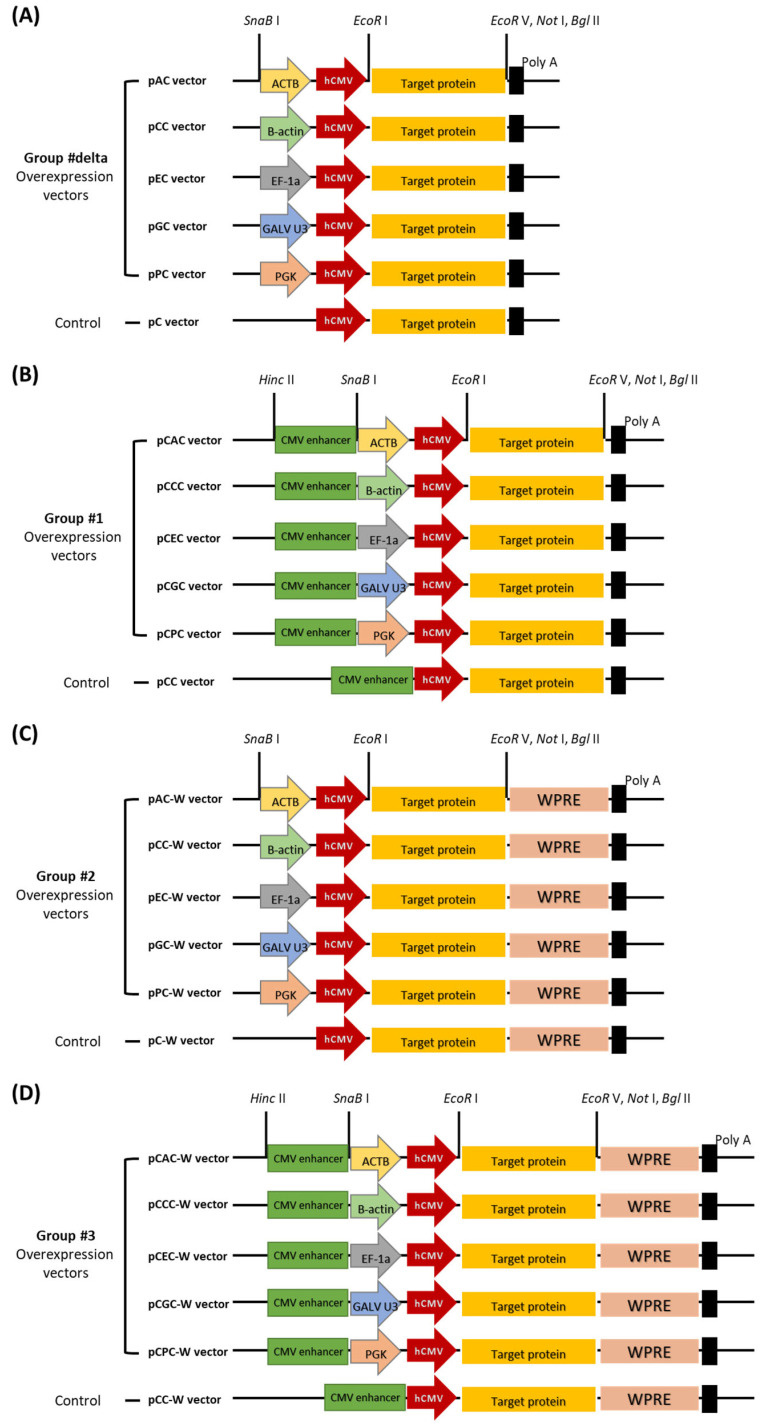
Final schematic diagram of overexpression vector construction according to placement of transcription and translation enhancers. The final schematic diagram for the construction of four groups of overexpression vectors was generated according to the additional arrangement of the transcription enhancer (cytomegalovirus enhancer) and translation enhancer (woodchuck hepatitis virus post-transcriptional regulatory element sequence). (**A**) Group lacking both transcription and translation enhancers. (**B**) Group with only transcription enhancer. (**C**) Group with only translation enhancer. (**D**) Group with both transcription and translation enhancers.

**Figure 2 ijms-25-02330-f002:**
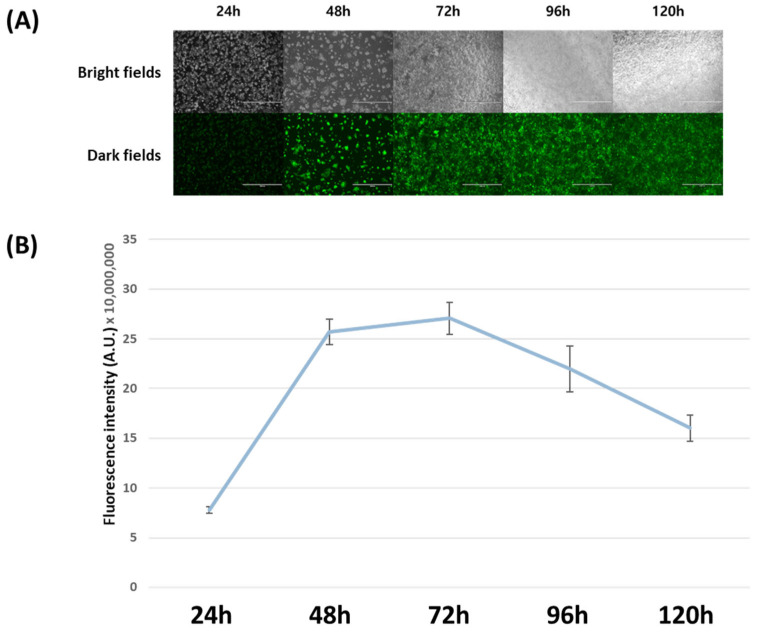
Confirmation of peak expression time post-transfection for relative comparison between overexpression vectors. Fluorescence micrographs (**A**) and intensities (**B**) of human embryonic kidney 293T cells transfected with overexpression vectors from 24 to 120 h post-transfection. The same field of vision was viewed in both bright fields and dark fields using a fluorescence microscope (Ts2R-FL, Nikon, Minato, Japan) (**A**) The scale bar indicates 1000 μm. The fluorescence intensity of the cell extracts was measured using a multimode microplate reader (SpectraMax i3x; Molecular Devices, San Jose, CA, USA), and is shown in arbitrary units (a.u.) (**B**). The bars indicate the means ± standard deviation (n = 3).

**Figure 3 ijms-25-02330-f003:**
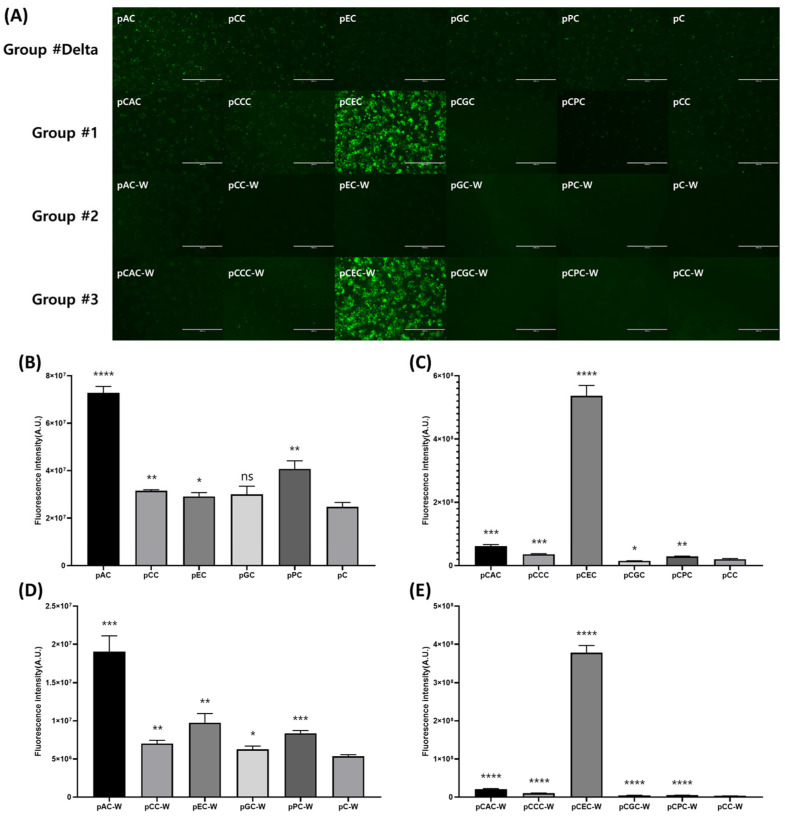
Fluorescence microscopy observation for relative comparison between overexpression vectors. Fluorescence micrographs (**A**) and intensities (**B**–**E**) of human embryonic kidney 293T cells transfected with overexpression vectors at 72 h post-transfection. The same field of vision was viewed in both bright fields and dark fields using a fluorescence microscope (**A**) The scale bar indicates 1000 μm. The fluorescence intensity of the cell extracts was measured using a multimode micro-plate reader, and is shown in arbitrary units (a.u.) (**B**–**E**). The bars indicate the means ± standard errors (n = 3) (ns, no significance; * *p* < 0.05, ** *p* < 0.005, *** *p* < 0.0005, **** *p* < 0.0001).

**Figure 4 ijms-25-02330-f004:**
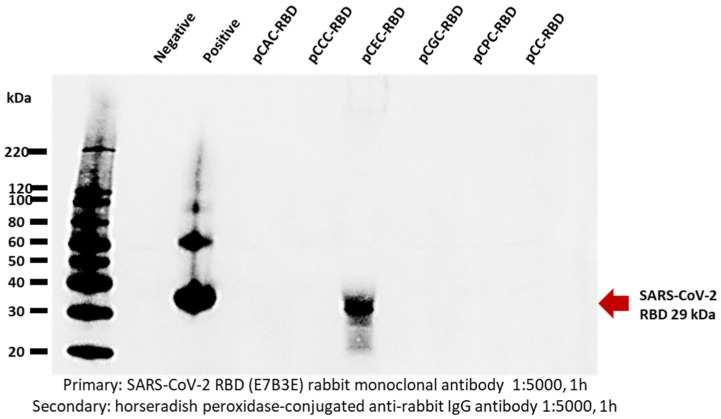
SARS-CoV-2 receptor-binding domain (RBD) protein expression by Group#1 overexpression vectors.

**Table 1 ijms-25-02330-t001:** Promoter, transcription enhancer, and translation enhancer for overexpression vector construction.

Expression Elements	Origin	Size (bp)	Application
CMV promoter	Human cytomegalovirus	204	The most widely used promoter: very strong gene expression promoter in most cellular systems.
Chicken B-actin promoter	Chicken	276	Strong expression promoter; highly efficient in stem cells.
Human EF1-a promoter	Human	1163	Strong expression promoter; highly efficient in stem cells.
PGK promoter	Mouse	500	Mouse phosphoglycerate kinase one promoter. Medium expression promoter.
GALV U3 promoter	Gibbon ape leukemia virus	322	GaLV envelope protein: enables gene transfer to various host cell domains of GALV and helps increase the cell infection rate.
ACTB promoter	Human	1505	Housekeeping genes: promoter with a CpG island that extends the proximal transcription start site (TSS).
CMV Enhancer	Human cytomegalovirus	380	Cis-regulatory acting element (CRE); immediate early gene transcription enhancer.
WPRE	Woodchuck hepatitis virus	688	Post-transcriptional regulatory element

EF1-a: elongation factor 1-alpha; PGK: phosphoglycerate kinase; GALV U3: gibbon ape leukemia virus U3; ACTB: actin, beta; CMV: cytomegalovirus; WPRE: woodchuck hepatitis virus post-transcriptional regulatory element.

## Data Availability

Data are available from the corresponding author upon request.

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
