# Peer review of "Novel Enhanced Mammalian Cell Transient Expression Vector via Promoter Combination"

_ijms, 2024, doi:10.3390/ijms25042330_

Round 1

Reviewer 1 Report

Comments and Suggestions for Authors

Introduction

There are several points in the introduction that should be improved:

·       The introduction seems cluttered with information without a clear direction or central theme. It discusses various aspects related to antigen proteins, vaccine development, expression systems, and vector construction without a clear transition or coherent flow.

·       The organization of ideas is not smooth. It jumps from discussing the importance of antigen proteins in vaccine development to the urgency posed by COVID-19, then shifts to discussing different vaccine platforms and expression systems. This lack of a smooth transition between ideas can confuse the reader.

·       Some sentences are quite long and convoluted, making it difficult to follow the argument. Breaking down complex sentences and rephrasing for clarity would greatly enhance readability.

·       Some statements, particularly those concerning the characteristics and limitations of various expression systems, lack specific citations. Adding references to support these claims would strengthen the credibility of the information presented.

·       The introduction repeats the importance of accurate efficacy evaluation and the necessity of the unique form of antigen protein expression multiple times, which can be redundant.

·       The transition from discussing the general importance of antigen proteins to the specific focus of the study (constructing a vector with increased transcriptional activity) is abrupt and could be made smoother.

·       While the introduction gives an overview of various challenges in antigen protein expression, it doesn’t clearly highlight how the study's approach or findings will address these challenges, aside from mentioning the construction of a vector with increased transcriptional activity.

Materials and methods:

Here are some areas of improvement in the Materials and Methods section:

·       The description of the methods lacks specific details about certain steps. For instance, the description of the construction of mammalian overexpression vectors is quite complex and might benefit from breaking down the process into more digestible steps.

·       The description of the vectors and groups might be confusing due to the complexity of the combinations. A clearer outline or schematic representation could aid in understanding the experimental design.

·       The sequence of methods could be more logically arranged. For instance, the description of plasmid transfection and fluorescence intensity measurement could precede the construction of overexpression vectors for better flow.

·       There's no mention of the statistical methods used for data analysis. Including information about statistical tests or analysis tools used would improve the rigor of the study.

·       Details about replicates, controls, and the number of times experiments were conducted are missing. Including this information is crucial for the reproducibility of the study.

·       The manufacturer's information for certain materials used in the study is vague. Providing full names or catalog numbers of reagents and equipment would improve reproducibility.

·       The description of the fluorescence intensity measurement process is intricate. Simplifying it or breaking it down into steps could enhance comprehension.

·       The Western blotting analysis is briefly described without specifying the quantification or analysis methods used. More detail on how the data were processed and quantified would be beneficial.

·       The duration for various experimental steps or intervals is not fully explained. Providing a clearer timeline for the experiments, especially for cell culture and sample collection, would add clarity.

Results:

This section presents several opportunities for improvement:

·       The use of technical terms might be overwhelming, and certain phrases, like "transcriptional or translational enhancers," are repeated frequently. Introducing these concepts in simpler terms or providing a glossary might aid comprehension.

·       The section lacks a smooth flow between descriptions, causing confusion. It transitions abruptly between the construction of vectors, protein expression time determination, comparative evaluations, and SARS-CoV-2 RBD protein expression. A clearer structure or segmentation would enhance readability.

·       There's a description of experimental results without clear conclusions or interpretations. Discussing the significance of the findings and their implications within the context of the study objectives would add depth.

·       Some critical details, such as the specifics of the optimal conditions for transfection or the rationale for selecting particular vectors, are absent. Including these details would help in understanding the experimental design.

·       Certain points, such as the comparison between different groups of vectors, are reiterated multiple times, which might be redundant. Streamlining the descriptions and focusing on essential points would enhance clarity.

Discussion:

Here are the areas that need improvement in the discussion section:

·       The discussion lacks a clear link to the study's objective and broader context. It jumps straight into specific findings without setting the stage or framing the results within the context of existing literature or the study's aim.

·       There's a heavy focus on describing the experimental outcomes without discussing their implications. This results in a descriptive rather than analytical approach, missing the opportunity to interpret or contextualize the results effectively.

·       The discussion highlights contradictory findings related to WPRE but lacks an in-depth exploration or potential explanations for these unexpected outcomes. Further explanations or hypotheses could improve the discussion's depth.

·       While some comparisons with previous studies are made, they lack depth and fail to contextualize how these findings contribute to or deviate from existing knowledge. More extensive comparisons to literature would add value.

·       The discussion lacks a detailed interpretation of the protein expression results, particularly regarding the failure to detect expression in certain vectors. A discussion on possible reasons for these failures beyond mRNA transcription would strengthen the analysis.

·       The section lacks statistical analyses or quantitative assessments to support the conclusions drawn. Incorporating statistical significance or confidence intervals for the differences observed in expression levels would increase the robustness of the findings.

·       While mentioning the potential of the developed vectors for rapid antigen provision, the discussion lacks detailed suggestions for future research directions or experiments to address unanswered questions.

·       The discussion reiterates the findings without exploring their significance or implications thoroughly. It's essential to move beyond reiterating results and delve into their importance in the field.

Comments on the Quality of English Language

Moderate editing of English language required

Reviewer 2 Report

Comments and Suggestions for Authors

The research article introduces a novel transient overexpression vector designed for the rapid production of large quantities of antigenic proteins in mammalian cell lines. Departing from exclusive reliance on the single human cytomegalovirus (CMV) promoter, the vector construction incorporates a combination of two promoters and includes a transcriptional enhancer (CMV enhancer), a translational enhancer (woodchuck hepatitis virus post-transcriptional regulatory element). Twenty vectors were created, with the human elongation factor 1-alpha (EF-1a) promoter-containing vector demonstrating the highest efficiency in expressing foreign proteins. This vector exhibited significantly elevated expression levels compared to the control vector, suggesting its potential for efficient and enhanced protein production, crucial for evaluating vaccine efficacy during emerging infectious diseases. The article highlights the importance of developing new overexpression vectors, emphasizing the synergistic effect achieved through the combination of promoters to enhance transcriptional activity.

In Section 4. Materials and Methods, it is essential to address the absence of information regarding the HEK293T cell line used. Given the heightened concern in the International Cell Line Authentication Committee (ICLAC) community, it is strongly recommended to furnish additional details about cell line sources and outline the strategies and frequencies of authentication tests, aligning with the recently published updated guidelines.

Concerning Figure 3, the observed very low level of fluorescence in most groups is noteworthy. While single CMV promoter constructs are typically effective for EGFP expression in vitro, the disparity should be elucidated to ensure the transparency and reliability of the results.

Figure 4 requires attention, as the Western Blotting image lacks a molecular ladder for comparison. Faint bands, except for the SARS-CoV-2 receptor-binding domain (RBD) protein, suggest potential issues. To establish the functionality of other constructs, it is crucial to include a molecular ladder, demonstrate Western Blots with higher protein loading and/or extended exposure time, and provide rationale for not testing constructs with WPRE elements.

In Line 43, the sentence stating, "This accelerated pace underscores the importance of promptly providing antigen proteins for evaluating the efficacy of newly developed vaccines upon the emergence of infectious diseases in the future" appears disconnected from the manuscript's focus on recombinant protein expression. It is advised to rephrase for clarity.

Line 50's statement about secretory or membrane proteins being "poorly expressed and difficult to mass-produce" is perplexing, as secretory proteins are generally considered easier to produce. Clarification is needed.

Regarding Figure 2, the term "Bright" should be corrected to "Bright field."

In Line 199, the phrase "... by enhancing stability and promoting processing" lacks specificity regarding the target of WPRE action. Authors should more accurately describe the mode of action of WPRE.

The discussion section necessitates improvement, specifically addressing possible mechanisms of dual promoter effects on transgene transcription. A more in-depth analysis of the literature and robust conclusions would enhance the overall quality of the discussion.

Details about the plasmid backbone used in the study are absent and should be provided for comprehensive understanding.

To fortify their results, authors should have performed qPCR measurements of EGFP and RBD mRNA levels, introducing a valuable dimension to their findings.

The generated results exhibit controversy and would benefit from additional controls and quality assurance steps to ensure the validity of the conclusions.

Comments on the Quality of English Language

Language is difficult to follow, authors make a lot of logical and factual mistakes or inaccuracies 

Reviewer 3 Report

Comments and Suggestions for Authors

Authors aimed to construct a vector with increased transcriptional activity in mammalian cell culture (the human embryonic kidney 293T (HEK293T) cell line).  In combination with the primary cytomegalovirus (CMV) promoter five auxiliary promoters (chicken B-actin, human elongation factor 1-alpha (EF-1a), phosphoglycerate kinase PGK, gibbon ape leukemia virus U3 GALV U3 and actin beta ACTB promoters) were used in the upstream region.  In addition, the influence of the CMV enhancer, in the upstream region, and woodchuck hepatitis virus post-transcriptional regulatory element (WPRE) in the downstream region of the target protein gene were studied. The level of expression of enhanced green fluorescent protein (EGFP) were estimated in 24 vector construction, whereas    expression of SARS-CoV-2 receptor-binding domain (RBD) protein were estimated in six the most promising construction. It was undeniably shown, that the vectors, employing the EF-1a as an additional promoter, exhibited the highest expression levels for both EGFP and SARS-CoV-2 RBD proteins. That can be attributed to the CMV enhancer, which stimulated both the EF-1a and CMV promoters and also enhanced  expression through the synergistic effect between them. Unlike the data of some other studies, in this study was shown, that WPRE inhibited the expression level of the overexpression  vector. This observation  is novel and was confirmed for the first time.  The manuscript is well written and illustrated and may be accepted as presented.

Round 2

Reviewer 1 Report

Comments and Suggestions for Authors

The author have been addressed most of comments

Comments on the Quality of English Language

Moderate editing of English language required especially the grammar side.

Reviewer 2 Report

Comments and Suggestions for Authors

All comments were addressed. Accept in present form